# Reviving a City's Economic Engine: The COVID-19 Pandemic Impact and the Private Sector's Engagement in Bandung City

Ahmad Zaini Miftah [1,*] , Ida Widianingsih [1,*] , Entang Adhy Muhtar [1] and Ridwan Sutriadi [2]

1 Public Administration Department, Faculty of Social and Political Sciences, Universitas Padjadjaran, Bandung 45363, Indonesia; entang@unpad.ac.id
2 Regional and Urban Planning Department, School of Architecture, Planning and Policy Development, Institut Teknologi Bandung, Bandung 40116, Indonesia; readone@sappk.itb.ac.id
* Correspondence: a.z.miftah@unpad.ac.id (A.Z.M.); ida.widianingsih@unpad.ac.id (I.W.)

**Abstract:** The COVID-19 pandemic has not only affected public health but has also significantly impacted the economy. Bandung, a bustling city in Indonesia serving as a satellite to the capital, has been hit hard due to its high population density, mobility, and reliance on the tourism, trade, and transportation sectors. Using a Computable General Equilibrium (CGE) model developed at the interregional level of Indonesia, this study investigates the microeconomic indicators of several economic activities in Bandung, namely, the transportation, accommodation and food–beverage, water supply, and trade (MSMEs) sectors, to assess the impact of the pandemic. Additionally, the study examines the role of private sector actors in contributing to the sustainable recovery efforts toward achieving the Sustainable Development Goals (SDGs) amidst the pandemic. The findings reveal that Bandung's transportation, accommodation, food and beverage, water supply, and trade sectors experienced a significant decline in economic activity. However, there was a gradual recovery, with increased economic activity between 2019 and 2021. Private sector actors and the health sector were the main drivers of economic recovery, with other sectors also contributing to the effort.

**Keywords:** economic activity; Computable General Equilibrium (CGE); Sustainable Development Goals (SDGs); post-pandemic recovery; policy integration; private sector engagement

## 1. Introduction

The COVID-19 pandemic has profoundly impacted the global community, posing a significant threat to public health and economic stability [1,2]. The pandemic has also had far-reaching effects on the progress toward achieving the 2030 Sustainable Development Goals (SDGs), triggering social, economic, and political changes that have resulted in development equity setbacks [3]. To curb the spread of the virus, many countries implemented measures such as population movement restrictions that slowed down economic activity, resulting in a trade-off between the health system and the economy [4–6]. The government was faced with the unique challenge of handling and minimizing risks associated with these trade-offs and developing integrated policies that involve various actors to build resilience and promote recovery after the pandemic [7–12].

The COVID-19 pandemic has also demonstrated that developed countries, despite their higher wealth, are not necessarily more resilient than developing countries [13,14]. Even several Western countries with advanced healthcare systems, such as France, England, Spain, Italy, and the United States, struggled to manage the pandemic, leading to many confirmed cases [15–17]. The impact of the pandemic is complex, posing distinctive challenges in its handling [18]. Governments face various uncertainties in policymaking due to unforeseen circumstances. This emphasizes the need for innovative and integrated policy management responses to health emergencies and their impacts [19,20].

In Indonesia, after the COVID-19 pandemic crisis was announced, the Indonesian government responded with Decision of the Ministry of Health No: HK.01.07/MENKES/104/2020

dated 4 February 2020 [21]. The COVID-19 pandemic has not only impacted the health system but has also had a significant impact on the economy. The effects of the pandemic were first seen in Q1 of 2020, with annual GDP growth at 3% (Figure 1), which continued to decline until the lifting of restrictions on population movement in Q4 of 2020 [22]. During this time, government spending was the only GDP component that showed significant positive growth, reflecting efforts to address the COVID-19 pandemic and support vulnerable groups [22]. The COVID-19 crisis decreased economic activity in sectors such as tourism [23–25], trade [26,27], and transportation [28,29].

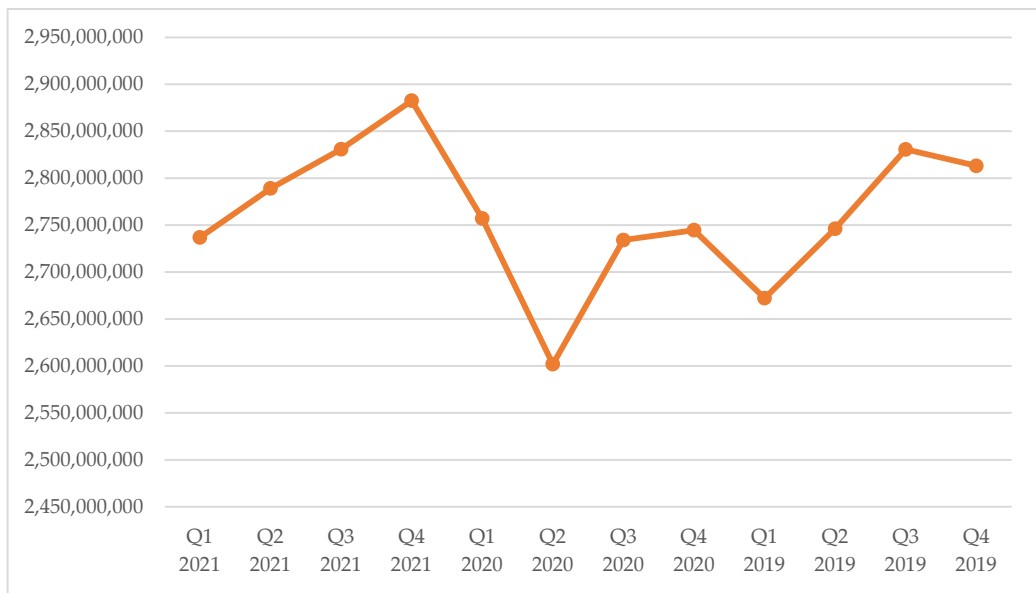

**Figure 1.** Indonesian GDP (Q1 2019–Q4 2020) (Source: https://www.bps.go.id/indicator/171/533/4/-seri-2010-2-pdrb-atas-dasar-harga-konstan-menurut-pengeluaran-2010-100-.html, accessed on 11 March 2020).

Bandung City is a densely populated and highly mobile satellite city of the nation's capital [30,31]. The city has long been a popular tourist destination due to its unique topography, historical buildings, cultural heritage, and delicious regional cuisine [32]. The city has adopted the branding "music-design-culinary" to promote cultural change and build a positive image as a tourism hub [32]. The Bandung City Government has implemented numerous policies and programs to develop micro, small, and medium-sized enterprises (MSMEs) in the trade sector [33]. The city has recently experienced rapid growth, as evidenced by the establishment of new buildings such as markets, malls, hotels, and other structures [34]. The transportation sector has also played a significant role in the city's growth, particularly in the tourism industry, with positive impacts on the local economy and surrounding areas that can help reduce poverty [35]. The city's Gross Regional Domestic Product (GRDP) data show that the trade, hotel, and restaurant sectors are the most significant contributors to the economy, accounting for 40.06% of the city's GRDP. The industrial sector contributed 25.73%, followed by the transport and communication sector (11.70%), service sector (9.15%), and the building sector (4.31%) [36].

The COVID-19 pandemic had a significant impact on the rate of economic growth in the city of Bandung. In 2020, the economic growth rate declined to −2.28% [37], a sharp drop from 6.79% in the previous year (Figure 2).

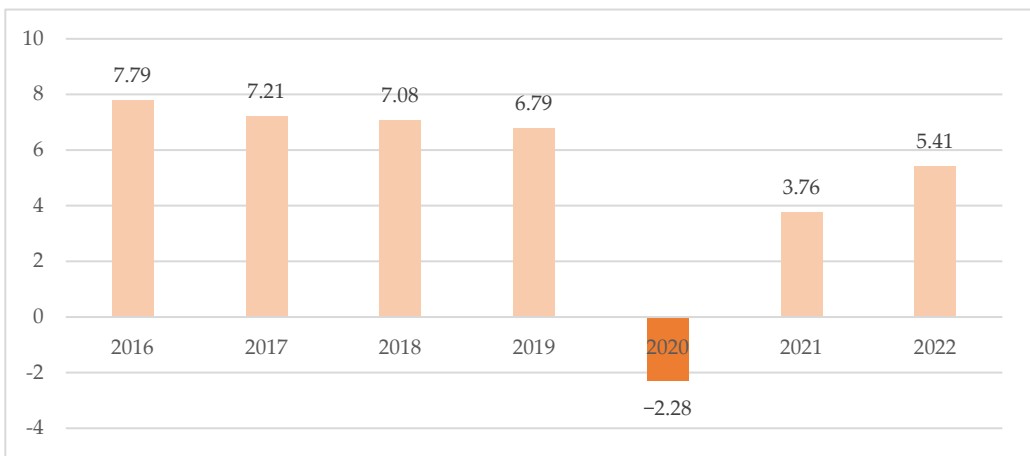

**Figure 2.** Bandung City economic growth rate (Source: https://bandungkota.bps.go.id/pressrelease/2022/02/25/971/pertumbuhan-ekonomi-kota-bandung-tahun-2021.html, accessed on 1 June 2023).

The decline in economic growth during the COVID-19 pandemic has highlighted the need for other sectors, such as NGOs, researchers, and communities, to play a role in the post-pandemic recovery [11,38,39]. Although some stakeholders sometimes have different perceptions regarding the handling of the COVID-19 pandemic, they are classified into three main groups, namely, those who respond positively, in a negative way and neutrally [40]. As a critical stakeholder in economic development, the private sector also plays a crucial role in economic recovery during a crisis. The coupling of public and private policies can mitigate the negative impact of economic activities on the SDGs [41]. This is particularly important when government resources are limited due to spending to deal with the COVID-19 pandemic [42].

Some studies investigate how the pandemic affects many aspects of life. One study aimed to provide a method for analyzing the variation in housing market demand caused by COVID-19 anti-contagion efforts in six metropolitan Italian cities [43]. Two property samples were assessed, one before and one after the implementation of COVID-19 measures. By identifying the most-valued features of residential real estate market demand, the proposed approach may assist public and private entities participating in urban investment decision-making. Another study assessed the influence of the COVID-19 pandemic on gender equality in terms of health, social, and economic indices [44]. We realize that it is also important to assess the economic impact on a local scale, particularly in the economic aspect, where Bandung, a bustling city in Indonesia that serves as a satellite to the capital, has been hard hit due to its high population density, mobility, and reliance on the tourism, trade, and transportation sectors.

Several studies have utilized the Computable General Equilibrium (CGE) model to analyze the impact of the COVID-19 pandemic. The CGE model is an effective tool for simulating economic scenarios [45]. For instance, Ref. [45] utilized the CGE model to assess the fiscal stimulus from the government that would offset the decline in GDP due to COVID-19. The study projected a decrease in GDP in most Brazilian states with sectoral projections at both the national and state levels. Another study conducted by [46] also employed the CGE model to evaluate the economic impact of transportation consumption during the COVID-19 pandemic in China. The researchers examined the impact of transportation investment policies as a government economic stimulus. The study results revealed that reduced transportation consumption affects China's macro-economy and decreases all industrial sectors' output. Furthermore, the service sector is the most adversely impacted. Based on these two studies, it is evident that gaining insight into the scale of the impact on several economic activity sectors at the local level in developing countries is crucial during a crisis. The objective of this study is to present an analysis of the impact of the COVID-19 pandemic on the local economic activity of Bandung City before, during, and

after the crisis, as well as to examine the role of private sector actors in addressing and recovering from the pandemic. It is crucial to assess the contributions of all sector actors to determine the sustainability of the economic recovery process, especially considering the crisis's impact on attaining the Sustainable Development Goals (SDGs).

## 2. Methods

This study employed the Computable General Equilibrium (CGE) method [47], specifically applied at the Indonesian inter-regional level statically. The study examined microeconomic indicators such as transportation, accommodation and food–beverage, water use, and trade (see Table 1).

**Table 1.** Observed indicators.

| GRDP Parameters | Observed Indicators | Data Used |
|---|---|---|
| Transportation and Warehousing | Transportation | Number of land transport passengers (public transportation) before, during, and after COVID-19 (P) |
| Provision of Accommodation and Food–Beverage | Accommodation and Food–Beverage | Hotel Tax Revenue before, during, and after COVID-19 (Ta) |
| | | Restaurant Tax Revenue before, during, and after COVID-19 (Tb) |
| Water Procurement, Waste Management, and Recycling | Water Procurement | Amount of Drinking Water Distributed in Bandung City (S) |
| Wholesale and Retail Trade; Car and Motorcycle Repair | Trade | Level of MSME Turnover in Bandung City (D) |

Source: Processed by authors (2023).

Below are the formulas and notations used in the static Computable General Equilibrium (CGE) modeling employed in this study.

1. Conditions before COVID-19 (assumed as 2019 conditions)

$$X = \sum_{i=1}^{N} C_i$$

$$C_i = P + T_a + T_b + S + D$$

2. Conditions during the COVID-19 Pandemic (assumed as the conditions in 2020 and 2021)

$$X' = \sum_{o=1}^{N} CO$$

$$C_0 = P + T_a + T_b + S + D$$

3. Conditions after the COVID-19 pandemic, i.e., the recovery stage (assumed as 2022 conditions)

$$X'' = \sum_{t=1}^{N} C_t$$

$$C_t = P + T_a + T_b + S + D$$

C = Conditions a, b, and c
X = Economic indicators observed in 2019 (baseline)
N = Total number of elements to be summed in a sequence
P = Number of land transport passengers before, during, and after COVID-19

Ta = Hotel Tax Revenue before, during, and after COVID-19
Tb = Restaurant Tax Revenue before, during, and after COVID-19
S = Amount of Drinking Water Distributed in Bandung City
D = Number of MSME Turnover in Bandung City.

To calculate the growth in economic activity, the following formula was used:

$$growth = \frac{\Delta_1 + \Delta_2 + \Delta_3}{X}$$

where $\Delta_1$ = economic activity growth 1 (2019–2020); $\Delta_2$ = economic activity growth 2 (2019–2021); $\Delta_3$ = economic activity growth 3 (2019–2022); and X = economic indicators observed in 2019 (baseline).

The economic assumptions utilized in analyzing the impact of the COVID-19 pandemic on the economic activities of the people of Bandung City via the CGE method were based on two main points. Firstly, the resulting impact on economic activity was observed only through economic indicators related to the transportation, accommodation and food–beverage, water procurement, and trade sectors. This is because these sectors were the most affected during the COVID-19 pandemic [11,38,48,49]. Secondly, post-pandemic data for COVID-19 were obtained through projections based on data trends from the previous year, as the status of the COVID-19 pandemic had only recently been revoked under the Instructions of the Minister of Home Affairs of the Republic of Indonesia Numbers 50 and 51 of 2022. The Health Emergency Status under RI Presidential Decree number 11 and 12 of 2020, as well as the Public Health Emergency of International Concern from WHO, are still being maintained by the Government of Indonesia in anticipation of a new wave of the COVID-19 pandemic.

While Data Exploratory Analysis [50,51] is used to gain insights from the roles and contributions of other actors (non-government actors), the analysis is carried out by exploring SER and CSR realization data for 2016–2021 provided by the Bandung City Planning, Research, and Development Agency in the fields of the environment, socio-economic factors, education, health, religious and spiritual activities, and sport, as well as the arts and culture, and then classifying them according to contributions that focus on efforts to deal with and recover from the COVID-19 pandemic.

## 3. Results

*COVID-19 Pandemic Impact on Economic Activities in Bandung City*

The COVID-19 pandemic was a public health crisis that has a multiplier effect on other sectors [52]. In Indonesia, social restriction policies have been implemented through Large-Scale Social Restrictions and the Enforcement of Restrictions on Community Activities [53]. Essentially, these policies limit population movement to curb the transmission of the COVID-19 virus, which impacts economic activity [54–56]. Transportation restrictions have a significant impact on economic activity as they disrupt distribution, value chains, consumption, and production, leading to an overall impact on development (Bonaccorsi et al., 2020; Martins et al., 2023; McKibbin & Fernando, 2020) [3,6,57].

The data indicate a significant decline in the number of passengers using public transportation (Trans Metro Bandung or TMB) during the COVID-19 pandemic population movement restriction policies (see Table 2). In 2018, the number of passengers was 958,453, which decreased to 990,873 in 2019, and further decreased to 360,749 in 2020 and 337,261 in 2021. However, the number of passengers increased in 2022 to 439,418. These data were converted into millions of rupiah units and projected for 2022 based on previous years' baseline data.

**Table 2.** Number of TMB passengers (P) before, during, and after the COVID-19 pandemic.

| TMB Passengers (P) in Million Rupiah | | | | | |
|---|---|---|---|---|---|
| Year | 2018 | 2019 | 2020 | 2021 | 2022 |
| Number of TMB Passengers | 958,453 | 990,873 | 360,749 | 337,261 | 439,418 |
| In million rupiah | 2157 | 2229 | 812 | 759 | 989 |

(Source: Processed by authors (2023) from http://satudata.bandung.go.id/, accessed on 1 June 2023).

Furthermore, Table 3 is the results of the hotel tax data processing before, during, and after the COVID-19 pandemic in Bandung City.

**Table 3.** Total Hotel Tax Revenue (Ta) Before, During, and After the COVID-19 Pandemic.

| Hotel Tax Receipt (Ta) in Million Rupiah | | | | | | |
|---|---|---|---|---|---|---|
| Year | 2017 | 2018 | 2019 | 2020 | 2021 | 2022 |
| Bandung Hotel Tax | 295,385 | 300,756 | 314,144 | 300,756 | 163,856 | 327,277 |

(Source: Processed by authors (2023) from http://data.jabarprov.go.id/, accessed on 1 June 2023).

The results of the restaurant tax data processing in Bandung City in million rupiahs are presented in the following Table 4.

**Table 4.** Total Restaurant Tax Revenue (Tb) Before, During, and After the COVID-19 Pandemic.

| Restaurant Tax Receipt (Tb) in Million Rupiah | | | | | | |
|---|---|---|---|---|---|---|
| Year | 2017 | 2018 | 2019 | 2020 | 2021 | 2022 |
| Bandung Restaurant Tax | 325,362 | 368,644 | 212,685 | 208,580 | 333,334 | 325,362 |

(Source: Processed by authors (2023) from http://data.jabarprov.go.id/, accessed on 1 June 2023).

Additionally, the following in Table 5 are the findings of the analysis of the consumption of drinking water in Bandung City. The data were compiled and then converted to millions of rupiahs by multiplying them by the average water tariff per cubic meter in Bandung City, which is Rp. 900/m$^3$. Moreover, projections for 2022 data were conducted using the baseline data from previous years.

**Table 5.** Amount of drinking water distributed in Bandung City (S) before, during, and after the COVID-19 pandemic.

| Amount of Drinking Water Distributed in Bandung City (S) | | | | | | | |
|---|---|---|---|---|---|---|---|
| | 2016 | 2017 | 2018 | 2019 | 2020 | 2021 | 2022 |
| Total (m$^3$) | 42,528,447 | 42,000,663 | 41,859,962 | 39,349,569 | 37,929,761 | 37,847,379 | 37,958,921 |
| In Million Rupiah | 38,276 | 37,801 | 37,674 | 35,415 | 34,137 | 34,063 | 34,163 |

(Source: Processed by authors (2023) from http://data.bandung.go.id/, accessed on 1 June 2023).

The following in Table 6 are the results of the processed data on the total turnover of each micro, small, and medium-sized enterprise (MSME) in Bandung City. Moreover, data projections have been conducted for 2021 and 2022 using baseline data from the previous several years.

**Table 6.** Total MSME turnover in Bandung City (D) before, during, and after the COVID-19 pandemic.

| Total MSME Turnover in Bandung City (D) in Million Rupiah | | | | | |
|---|---|---|---|---|---|
| | 2018 | 2019 | 2020 | 2021 | 2022 |
| In Million Rupiah | 508,665 | 608,808 | 614,787 | 561,726 | 661,869 |

(Source: Processed by authors (2023) from http://satudata.bandung.go.id/, accessed on 1 June 2023).

The CGE method is crucial in evaluating the effects of major shocks, such as the COVID-19 pandemic crisis, on regional economic activity [47]. Below in Tables 7 and 8 are the results of an analysis of the impact of the COVID-19 pandemic on economic activity in Bandung City.

**Table 7.** CGE calculations on observed economic activities in Bandung City.

| | P | Ta | Tb | S | D | $C_i$ | X |
|---|---|---|---|---|---|---|---|
| **Before the COVID-19 Pandemic (2019) in Million Rupiah** | | | | | | | |
| 2019 | 2229 | 314,143 | 368,643 | 35,414 | 608,807 | 1,329,239 | 1,329,239 |
| **During the COVID-19 Pandemic (2020 and 2021) in Million Rupiah** | | | | | | | |
| | P | Ta | Tb | S | D | $C_0$ | X' |
| 2020 | 812 | 300,756 | 212,685 | 34,137 | 614,787 | 1,163,176 | 1,163,176 |
| 2021 | 759 | 163,856 | 208,580 | 34,063 | 561,726 | 968,985 | 968,985 |
| **After the COVID-19 Pandemic (2022) in Million Rupiah** | | | | | | | |
| | P | Ta | Tb | S | D | $C_t$ | X'' |
| 2022 | 989 | 327,277 | 333,334 | 34,163 | 661,869 | 1,357,631 | 1,357,631 |

**Table 8.** Growth of economic activity generated from CGE analysis.

| Economic Activity Impact | | | |
|---|---|---|---|
| Δ1 | Δ2 | Δ3 | Growth |
| −166,064 | −360,255 | 28,391 | −37.41% |
| −12.49% | −27.10% | 2.14% | |

Based on the results presented above, Delta 1 (Δ1) reflects the changes in economic activity during the COVID-19 pandemic in 2020 compared to 2019, before the pandemic affected Bandung City. This means there was a decrease in economic activity in the transportation, accommodation and food–beverage, water supply, and trade (MSME) sectors by −12.49% in 2020, which can be attributed to the impact of the COVID-19 pandemic. The negative number indicates a decline in several economic activities, which can be considered a significant negative impact of the COVID-19 pandemic on economic activity in Bandung City.

Delta 2 (Δ2) represents the changes in economic activity during the COVID-19 pandemic in 2021 compared to 2019, before the pandemic affected Bandung City. This indicates a decrease in economic activity in the transportation, accommodation and food–beverage, water supply, and trade (MSME) sectors by 27.10% in 2021, which can be attributed to the impact of the COVID-19 pandemic. The negative value implies a substantial negative impact of the pandemic on several economic activities in Bandung City.

Delta 3 (Δ3) illustrates changes in economic activity after the peak wave of COVID-19, namely in 2022, compared to 2019 before the arrival of COVID-19 in Bandung City. This indicates that there has been growth in economic activity in the transportation, accommodation and food–beverage, water supply, and trade (MSME) sectors of 2.14% in 2022, which can be attributed to the impact of the COVID-19 pandemic. This positive number suggests that there has been an increase in several economic activities in Bandung City. Consequently, in 2022, Bandung City has shown signs of recovery, with a notable increase of 2.14% in several sectors of economic activity.

Growth indicates the change in overall economic activity in the region due to the COVID-19 pandemic. In this case, the growth value of the economic activity in the transportation, accommodation and food–beverage, water supply, and trade (MSME) sectors is −37.41%. This indicates a decrease in the value of overall economic activity by 37.41%

compared to the value of economic activity before the COVID-19 pandemic. This represents a significant negative impact on the growth of economic activity in Bandung City [3,10,11,38,42,58–62].

## 4. Discussion

### *4.1. COVID-19 Pandemic Impact on Economic Activities in Bandung City*

The COVID-19 pandemic has directly impacted the efforts to achieve the 2030 Sustainable Development Goals (SDGs) due to slowed economic growth and weakened purchasing power resulting from mobility restrictions aimed at suppressing the transmission of the COVID-19 virus. Furthermore, it has raised the threat of a global recession. During the pandemic, the economic growth rate in Bandung City declined significantly, reaching −2.28%. The pandemic shock affected several sectors, including transportation, tourism, and others, causing a slowdown in economic activity before, during, and after the pandemic. Data from the CGE analysis shows a significant decline in economic activity of 12.49% before the pandemic in the transportation, accommodation and food–beverage, water supply, and trade sectors in Bandung City. Economic activity decreased by 27.10% during the pandemic, but when the situation started to improve and plateaued, there was a 2.14% increase in economic activity. This demonstrates the significant impact of the pandemic on economic activity in Bandung City. Based on these data, when there is a trade-off between the health and economic sectors, the health sector must take precedence in COVID-19 to provide opportunities to increase people's economic activities.

The CGE analysis results given demonstrate the trajectory of the changes in economic activity in Bandung City during the COVID-19 pandemic (see Figure 3). The decline that occurred in Delta 1 and Delta 2 shows the general understanding that these sectors are highly vulnerable to restrictions and decreased demand during the pandemic. An interesting change occurred in Delta 3, where there was a positive change in economic activity in 2022 compared to 2019. This growth of 2.14% shows signs of recovery and increased economic activity after passing through the peak wave of the COVID-19 pandemic.

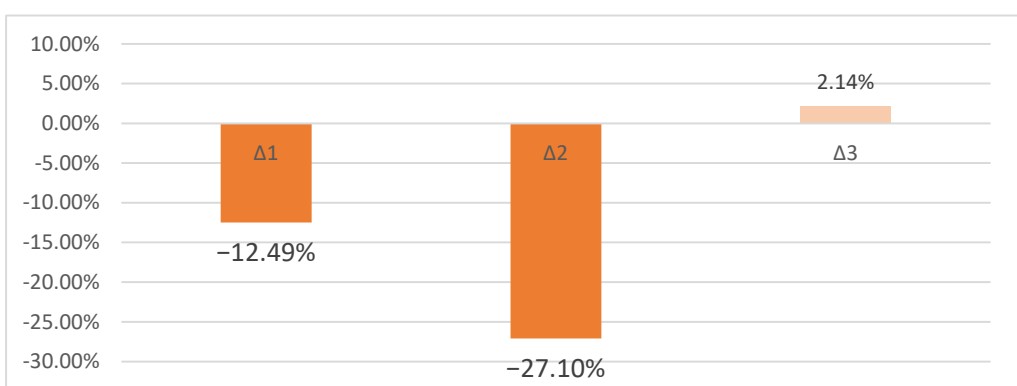

**Figure 3.** The trajectory of changes in economic activity in Bandung City during the COVID-19 pan-demic.

Economic activity experienced positive changes in 2022, showing signs of recovery and improvement after going through the difficult times of the pandemic. This can be interpreted as a result of efforts to deal with the COVID-19 pandemic, including economic recovery efforts and the easing of restrictions by the government and businesses. However, it should be noted that, while there is growth in Delta 3, it is still limited and has yet to fully restore economic activity to pre-pandemic levels. The significant decline in overall growth (37.41%) shown in the CGE analysis shows that the impact of the pandemic on Bandung City's economic activities is still very significant. Despite the recovery in 2022, the significant decline and changes in growth underscore the importance of understanding the economic consequences of the pandemic and the need for appropriate recovery measures. Although growth occurred in Delta 3 or in 2022, this growth was still limited and has not been able to

restore the economic activity of Bandung City to pre-pandemic levels. This shows that the impact of the pandemic on economic activity is still ongoing and has a significant impact.

### 4.2. Private Sector Involvement in the COVID-19 Pandemic Handling and Recovery in the City of Bandung

The COVID-19 pandemic poses challenges not only for the health sector but also for the economic, social, political, and environmental sectors, and may lead to a global recession due to the cessation of mobility and slowing economic growth [11,38]. Moreover, conversely, chaos in the social context, economic, and political environment also played a role in creating the perfect storm of crises [58]. COVID-19 has revealed a different way of life from usual [59]. Moreover, the pandemic has jeopardized sustainable development achievements [3]. Even without the pandemic, achieving the 2030 sustainable development targets would be challenging for several countries [38]. To attain sustainable development, it is essential to focus on economic development and other aspects, such as social culture and the environment [60].

The recovery from the COVID-19 pandemic, under the slogan "Build Back Better", presents an opportunity and a challenge for fostering innovation in sustainable development [11,61]. Achieving sustainable development is not an easy task and requires the involvement and contribution of various actors based on the Sextuple Helix Model, ranging from local government to the international community [10,11]. Policy integration, both vertically and horizontally, is also crucial to support the effective implementation of sustainable development [62]. The government needs to learn and concretize the role of policy integration from the complexity of handling the COVID-19 pandemic for coordinating multi-sectoral policies toward achieving the Sustainable Development Goals [10].

The role of local authorities in developing an integrative strategy that involves stakeholders such as the business sector, NGOs, researchers, and other communities is crucial for post-pandemic recovery from COVID-19 [10,11,38]. The private sector's contribution is essential, mainly when government resources are limited due to the spending on handling the COVID-19 pandemic [42]. In handling and recovering from the COVID-19 pandemic in Bandung City, private companies, regionally owned enterprises, and state-owned enterprises have contributed through two programs, namely Social and Environmental Responsibility (SER) and Corporate Social Responsibility (CSR). A number of companies in Bandung City have offered aid and resources in the form of contributions, medical equipment purchase, support for disadvantaged groups, and other activities to assist communities in dealing with the economic burden of the pandemic (see Figures 4 and 5) [63].

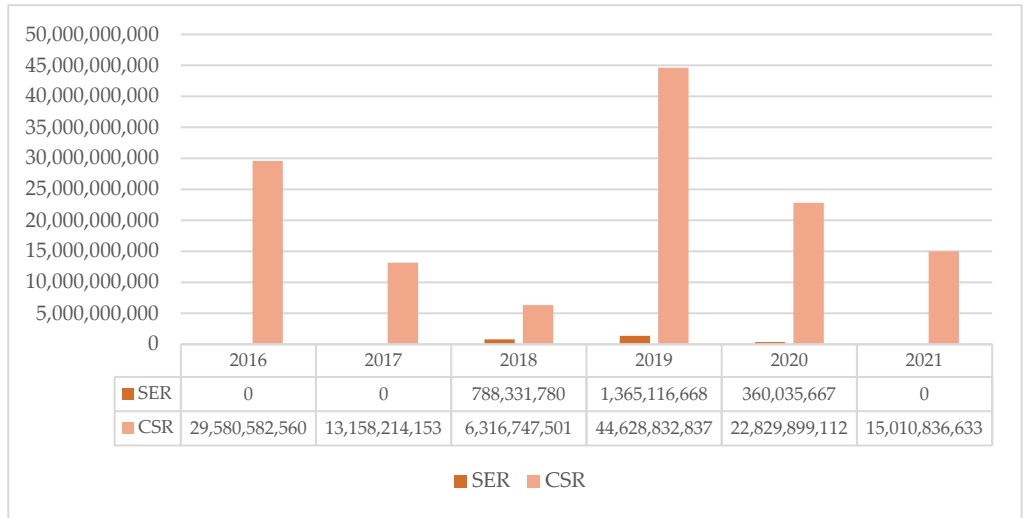

**Figure 4.** Level of post-pandemic handling and recovery assistance through the SER and CSR programs (Rp) (*Source: Processed from* Performance Report 2020 Social and Environmental Responsibility Forum (TJSL) Period 2018-2022 http://tjsl.bandung.go.id/about-us, accessed on 12 March 2022).

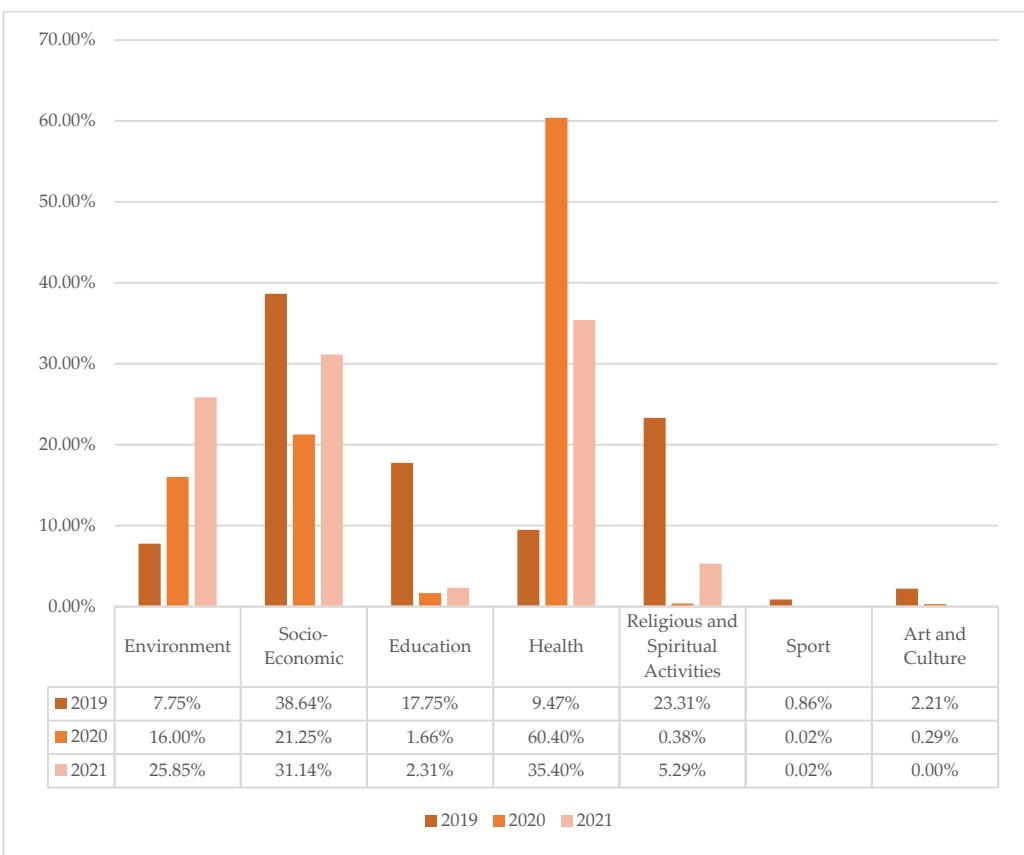

**Figure 5.** Percentage of per sector allocation of post-pandemic handling and recovery assistance through SER and CSR programs (Source: Processed from Performance Report 2020 Social and Environmental Responsibility Forum (TJSL) Period 2018–2022 http://tjsl.bandung.go.id/about-us, accessed on 12 March 2022)).

In 2020, private sector assistance increased significantly by 1.3 billion from SER and 44.6 billion from CSR funds, despite strict social restrictions imposed in Bandung City, leading to a slowdown in economic growth to −2.28% in 2020 [37]. Private sector assistance focused on providing basic needs and social safety nets to stabilize society. Additionally, private sector assistance facilitated the health sector by providing oxygen cylinders and COVID-19 kits and implementing education and law enforcement programs to curb virus transmission. The most significant proportion of private sector contributions for handling and recovering after the COVID-19 pandemic in Bandung City from 2019 to 2021 was in the economic recovery sector at 32.52%, followed by the health sector at 28.12%, the religious sector at 13.25%, the environmental sector at 13.25%, the education sector at 10.57%, the arts and culture sector at 1.29%, and the sports sector at 0.48%. Various industrial fields and types of companies, including private companies, regionally owned enterprises, and state-owned enterprises, were significant contributors outside the Bandung City Government through SER and CSR programs (see Figure 6). The size of the circle in the illustration represents the percentage of private-sector contributions used in the COVID-19 pandemic handling and recovery.

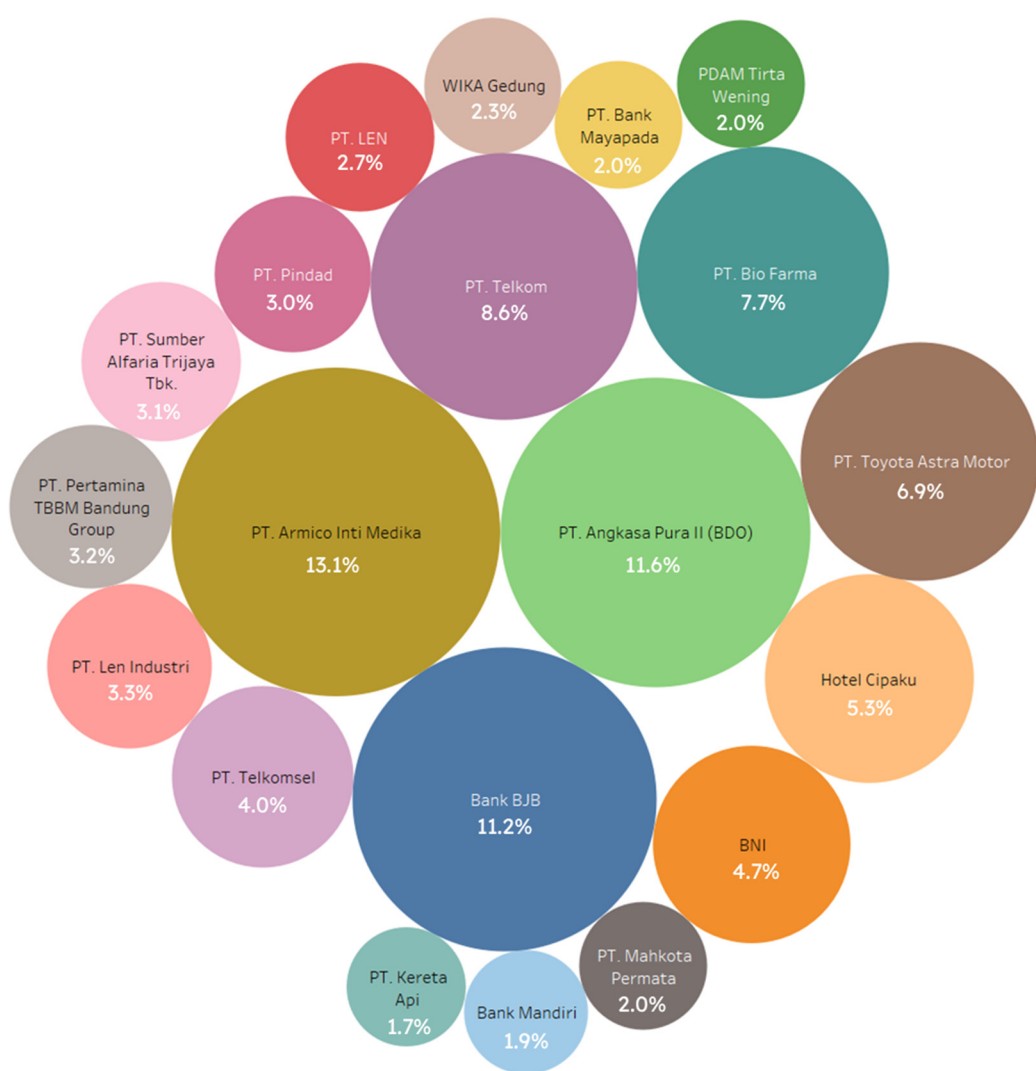

**Figure 6.** The 20 most dominant actors contributing to the handling and recovery of the impact of the COVID-19 pandemic through SER and CSR programs in Bandung City (2019–2021).

The impact of the COVID-19 pandemic is complex. It has created a multiplier effect, requiring an integrative strategy that involves various actors through vertical and horizontal policy integration; the recovery efforts from the COVID-19 pandemic present opportunities and challenges in creating sustainable development innovations. When government resources are limited, the private sector plays a critical role in handling and recovering from the pandemic. The Bandung City Government involves the private sector through Social and Environmental Responsibility (SER) and Corporate Social Responsibility (CSR) programs to deal with and overcome the COVID-19 pandemic. By combining private and public sector efforts, the impact of changes in economic activity caused by the pandemic can be mitigated to achieve the Sustainable Development Goals. Furthermore, the complexity of the challenges posed by the pandemic demonstrates that the integration process requires a shared commitment to achieving sustainable development objectives.

We can see that the SDGs encompass a wide range of development issues, including social, economic, and environmental concerns. The social and economic constraints put in place to limit the virus's spread resulted in a slowdown in economic activity and lower economic growth, and impeded progress toward the SDGs. CSR and SER are the terms used to describe the notion of corporations actively contributing to sustainable development and accepting responsibility for the social, economic, and environmental repercussions of

their activities. These private sector organizations play a crucial role in assisting impacted people and the environment in the context of the COVID-19 pandemic.

## 5. Conclusions

The significant decline in overall growth highlights the magnitude of the pandemic's impact on Bandung City's overall economic activity. This result reflects a substantial decline in economic activity and emphasizes the importance of understanding and addressing the economic consequences of the pandemic. In this context, the COVID-19 pandemic has had a significant and sustained impact on economic activity in Bandung City. There is a need for appropriate and sustainable recovery measures to address the impacts of the pandemic. Such measures could involve policies that support the most affected sectors of economic activity, comprehensive economic recovery efforts, and enhancing the adaptability and resilience of Bandung's economy to future challenges. It is also important to have a deeper understanding of the economic consequences of the pandemic. A more comprehensive and holistic analysis, including an understanding of the social, cultural, and environmental impacts, is necessary to fully understand the effects of the COVID-19 pandemic and design appropriate recovery strategies. As such, this analysis underscores the need for a comprehensive, critical, and sustainable response to the impact of the COVID-19 pandemic on economic activities in Bandung City.

The role and contribution of the private sector in the city of Bandung are significant during the handling and recovery of the COVID-19 pandemic. The private sector was the most significant contributor to economic recovery in Bandung during the 2019–2021 period, followed by the health sector, and the religious, environmental, and educational sectors. The arts, culture, and sports sectors made relatively minor contributions. Therefore, economic recovery in Bandung during the 2019–2021 period was mostly driven by the private and health sectors, while other sectors also contributed. Achieving the 2030 Sustainable Development Goals (SDGs) in the aftermath of the COVID-19 pandemic requires contributions from multiple stakeholders (SDG 17—Partnerships for the Goals) with integrative strategies. By combining efforts from both the private and public sectors, the impact of changes in economic activity caused by the pandemic can be mitigated to achieve the Sustainable Development Goals. From a policy integration perspective, the complexity of the challenges brought about by the pandemic serves as a lesson that the integration process requires a shared commitment to achieving the Sustainable Development Goals.

It is important to note that this study is limited in its focus on microeconomic indicators, specifically transportation, accommodation and food–beverage, water usage, and trade. It describes only the magnitude of the private sector's contribution during the handling and recovery from the COVID-19 pandemic. Further studies are needed to explore other economic indicators and the motives for actors' involvement.

**Author Contributions:** Conceptualization, I.W. and A.Z.M.; methodology, I.W. and A.Z.M.; software, A.Z.M.; validation, I.W., E.A.M. and R.S.; formal analysis, I.W., E.A.M. and R.S.; investigation, I.W. and A.Z.M.; resources, I.W. and A.Z.M.; data curation, I.W., E.A.M. and R.S.; writing—original draft preparation, I.W. and A.Z.M.; writing—review and editing, A.Z.M., I.W., E.A.M. and R.S.; visualization, A.Z.M.; supervision, I.W. All authors have read and agreed to the published version of the manuscript.

**Funding:** The APC was funded by Universitas Padjadjaran.

**Institutional Review Board Statement:** Not applicable.

**Informed Consent Statement:** Not applicable.

**Acknowledgments:** The author would like to thank all researchers from the Bandung City Planning, Research, and Development Agency and Quadran Energi Rekayasa, AP Management and Policy Studies who have supported this study.

**Conflicts of Interest:** The authors declare no conflict of interest.

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
