# Peer review of "Reviving a City’s Economic Engine: The COVID-19 Pandemic Impact and the Private Sector’s Engagement in Bandung City"

_sustainability, doi:10.3390/su15129279_

Round 1

Reviewer 1 Report

The article is devoted to the topical topic of economic recovery and development during the covid-19 pandemic and the post-pandemic period. The authors used a proven model used in Indonesia to determine the impact of the pandemic on economic dynamics. Notes to the manuscript. Firstly, what immediately catches the eye is a very short Discussion section and the absence of a Conclusion section.  Paragraph 3..2 is more suitable for the Discussion section, while the Discussion section is more relevant to Conclusions. It is probably necessary to take this into account for the authors to rework the article. In general, the figures and tables are clear and indicative, but it would be more presentable to perform them in color. The article can be accepted after correcting the text according to the comments mentioned above.

Author Response

Dear Reviewer,

Thank you for your feedback on the article. We appreciate your insightful observations regarding the Discussion and Conclusion sections. We would like to inform you that we have taken your suggestions into account and made the necessary changes to improve the structure and clarity of the article.

Firstly, we have included a comprehensive Conclusion section to provide a concise of the key findings and implications of the study. 

Additionally, we have repositioned Paragraph 3.2 to the Discussion section, as you rightly pointed out its suitability for that section. This adjustment ensures that the content is appropriately categorized and presented for a better understanding of the study's results and their interpretation.

Furthermore, we have revamped the visual elements, such as tables and figures, by incorporating a color scheme. This modification aims to enhance the visual appeal and improve the overall readability of the article.

Once again, we sincerely appreciate your valuable feedback, and we believe that these revisions have addressed the concerns you raised. We are confident that these changes have strengthened the article and improved its overall quality.

Reviewer 2 Report

Please see the attached review.

Please see the attached review. 

Author Response

Dear Reviewer,

Thank you for your feedback on the article. We want to assure you that we have carefully considered all of your suggestions and have made efforts to address and improve the areas you highlighted. We appreciate your valuable input, as it has helped us enhance the overall quality and clarity of the article. We are confident that the revisions we have made will meet your expectations and make the article more valuable to its readers.

Reviewer 3 Report

The aim of the work is to study the microeconomic indicators of several economic activities in Bandung to assess the impact of the pandemic.

Some suggestions are following provided in order to improve some weaknesses of the research:

Abstract

The microeconomic indicators mentioned can be esplicitate in order to know the issues that the Authors are studing.

Introduction

The literature review section is quite missing, if it is possible, it could be fine to add more references that have addressed the same topic or have used the same methodology/indicators system to analyse the covid-19 effects. Some suggestions could be: Tajani, F., Liddo, F. D., Guarini, M. R., Ranieri, R., & Anelli, D. (2021). An assessment methodology for the evaluation of the impacts of the COVID-19 pandemic on the Italian housing market demand. Buildings, 11(12), 592 and Flor, L. S., Friedman, J., Spencer, C. N., Cagney, J., Arrieta, A., Herbert, M. E., ... & Gakidou, E. (2022). Quantifying the effects of the COVID-19 pandemic on gender equality on health, social, and economic indicators: a comprehensive review of data from March, 2020, to September, 2021. The Lancet, 399(10344), 2381-2397.

Methods

In table 1 the data source and the reference years are missing

Figure 6 should be improved

Conclusions

More discussion of the results is required, from a critical point of view. Moreover, the future developments and the practical implications need to be improved.

Author Response

Dear reviewer,

Thank you for your feedback on the article. We want to assure you that we have carefully considered all of your suggestions and have made efforts to address and improve the areas you highlighted. We appreciate your valuable input, as it has helped us enhance the overall quality and clarity of the article. We are confident that the revisions we have made will meet your expectations and make the article more valuable to its readers.

Round 2

Reviewer 2 Report

I don't have any specific comment/suggestion.

The current version is much improved. 

Reviewer 3 Report

The efforts made by the Authors are apprecciated